# Calcium Channel Blocker-Related Chylous Ascites: A Systematic Review and Meta-Analysis

**DOI:** 10.3390/jcm8040466

**Published:** 2019-04-05

**Authors:** Meng-Ko Tsai, Chao-Hung Lai, Li-Mien Chen, Gwo-Ping Jong

**Affiliations:** 1Department of Internal Medicine, Taichung Armed Forces General Hospital, Taichung 41168, Taiwan; Raymond-620@hotmail.com (M.-K.T.); sghandsw@yahoo.com.tw (C.-H.L.); 2National Defense Medical Center, Taipei 11490, Taiwan; 3Department of Internal Medicine, Chung Shan Medical University Hospital and Chung Shan Medical University, Taichung 40201, Taiwan

**Keywords:** chylous ascites, calcium channel blockers, milky ascites/effluents, end-stage renal disease

## Abstract

Background: Chylous ascites is an uncommon condition characterized by a white, milky-appearing peritoneal fluid, and is related to disruption of the lymphatic system from any cause. There have been very few previous reports of calcium channel blockers (CCBs) as potential causes of chylous ascites, and most of the patients were undergoing peritoneal dialysis. Aims: To review the pathogenesis, clinical manifestations, laboratory examinations, treatment options, and prognosis of patients with CCB-related chylous ascites. Method: A retrospective analysis was conducted for patients with CCB-related chylous ascites from publications in PubMed, EMBASE, and LILACS between January 1993 and December 2018. Results: A total of 48 cases were included. The average age at disease onset was 50.2 ± 10.9 years, with a male:female ratio of 1.5:1.0. The symptoms of abdominal distension/pain and chylous ascites were observed within 48–72 h of drug initiation and disappeared within 24 h of drug withdrawal. Rechallenge was performed in 10 patients, and all (100%) of them showed chylous effluents that disappeared within 24 h after stopping drug treatment. Conclusions: To summarize, CCB-related chylous ascites is formed of white, milky ascites/effluents that appear after administration of CCBs. Physicians must be aware of the possibility of chylous ascites when administering CCBs, particularly in patients with renal function impairment or patients with end-stage renal disease who are undergoing peritoneal dialysis.

## 1. Introduction

Chylous ascites, which is defined as the accumulation of triglyceride-rich fluid in the peritoneal cavity, is an uncommon condition characterized by a white, milky-appearing peritoneal fluid and is related to disruption of the lymphatic system from any cause. Any source of lymphatic vessel obstruction or leakage can potentially cause chylous ascites [1,2], and it is generally associated with malignancies and surgical trauma [2]. There have been very few previous reports regarding chylous ascites arising secondary to administration of drugs [3].

Calcium channel blockers (CCBs) are widely used for the treatment of hypertension. There have been few reports of CCBs as potential causes of chylous ascites [3], therefore, information on this topic is limited. The aim of this study was to review the literature concerning CCB-related chylous ascites to outline the pathogenesis, clinical manifestations, laboratory examinations, treatment options, and prognosis of this rare disease entity from January 1993 to December 2018.

## 2. Methodology

### 2.1. Literature Search and Screening

A systematic article search using electronic databases was conducted by 2 independent medical doctors (C.-L.M. and L.-G.H.). If there was an inconsistent selection and lack of agreement, another senior medical doctor (J.-G.P.) made the final judgment and decision. The electronic databases of MEDLINE (PubMed), EMBASE, and LILACS were searched from January 1993 to December 2018. The medical subject heading terms of chylous ascites, chyloperitoneum, cloudy/turbid dialysate or effluent, and CCBs were used. The search was limited to humans and published in English.

### 2.2. Clinical Data

All patients with CCB-related chylous ascites were included in the study. The patients’ sex, age, country, whether they had undergone hemodialysis, clinical findings, time for the chylous ascites/effluent to develop and subside, serum triglyceride (TG) levels of ascites/effluent, and whether a rechallenge test of the CCBs was performed were recorded.

### 2.3. Statistical Methods

Continuous variables were expressed as valid percentages and mean values with the standard deviation (SD). For univariate analyses, *t*-tests were performed. Categorical variables were expressed as frequency. Differences were assessed by the chi-squared test or Fisher’s exact test. A *p* value of <0.05 was considered as indicative of statistical significance. SPSS 19.0 statistical software (SPSS, Chicago, IL, USA) was used to perform all statistical calculations. 

## 3. Results

### 3.1. Clinical and Laboratory Findings of All Patients

From January 1993 to December 2018, 13 articles [2,3,4,5,6,7,8,9,10,11,12,13,14] and 47 cases were found, and together with our one case, a total of 48 cases were included in the study. The available data revealed 22 males and 23 females (the sex of three patients was not known), with a median age of 50 years. Forty-five patients received peritoneal dialysis (PD) because of end-stage renal disease (ESRD). Three patients were noted to have renal function impairment and had not received PD (Table 1).

### 3.2. Chylous Ascites Development/Subsidence

The time to develop chylous ascites/effluent after taking CCBs had been recorded for 47 patients. Chylous ascites/effluent developed within 24 h in nine (19.1%) of them, within 48 h in 34 (72.3%), and within 72 h in 38 (80.1%). Associated symptoms/signs were recorded for 44 (97.8%) of the patients whose chylous ascites/effluent subsided shortly/rapidly within 24 h after ceasing the CCBs. Ten patients were rechallenged with CCBs, and all (100%) again developed chylous effluents that disappeared after stopping the CCBs treatment (Table 1 and Table 2). Thirty-three patients showed clinical manifestations, nine (27.3%) of them had abdominal distension/pain, and the other 24 (72.7%) had no abdominal discomfort (Table 2).

### 3.3. TG (Chylous Ascites/Effluent) Levels

Twenty-nine patients had chylous ascites/effluent TG levels. Two of the 29 had renal function impairment but had not undergone PD, and 27 of the 29 had received PD because of ESRD. The mean chylous ascites TG level in two of three patients who had renal function impairment but had not undergone PD was 201 mg/dL. The mean effluent TG level of the 27 patients who had received PD was 74.1 mg/dL. There were significant differences in the mean effluent TG level between the patients who had received PD and had not received PD (*p* < 0.01). The remaining mean effluent TG level of 17 patients from three retrospective/prospective studies was 70.6 mg/dL (Table 3).

### 3.4. Lipophilicity

There may be a relationship between lipophilicity and CCB-associated chylous ascites. The XlogP3 values of CCB-associated chylous ascites are listed in order of lipophilicity in Table 1. The partition-coefficient (log *P*) is defined as the ratio of the concentrations of a compound in a mixture of the aqueous phase and of an un-ionized compound in an immiscible lipophilic phase at equilibrium. A high log *P* indicates a preference toward hydrophobic interactions, which is interpreted as lipophilicity [15]. There are many methods for estimating log *P*, and XlogP3 values almost perfectly correlate with log *P* values [16]. The XlogP3 values of compounds are available in the PubChem database [17], and the XlogP3 values of CCBs are listed in Table 4.

Table 4 shows the reported CCBs found to be related to chylous ascites. Twenty-five (52.1%) of the total 48 patients had chylous ascites caused by lercanidipine with a lipophilicity of 6.9, followed by manidipine with a lipophilicity of 5.6.

## 4. Discussion

Chylous ascites is a condition in which white milky fluids rich in triglycerides (usually >200 mg/dL) accumulate in the abdominal cavity [2]. The characteristics of the ascitic fluid in chylous ascites include triglycerides >200 mg/dL, above the 500 leucocyte count, between 2.5 and 7.0 g/dL total protein, below 1.1 g/dL serum-ascites albumin gradient (elevated above 1.1 g/dL in chylous ascites secondary to cirrhosis), low total cholesterol (ascites/serum ratio <1), between 110 and 200 IU/L lactate dehydrogenase, positive culture in selected cases of tuberculosis, positive cytology in malignancy, elevated amylase in cases of pancreatitis, and below 100 mg/dL glucose. There are multiple causes, which used to be divided into primary and secondary [18]. Primary chylous ascites is often caused by a congenital lymphatic abnormality. Secondary chylous ascites, which is by far more common, is usually caused by malignancy (principally lymphoma), surgery, or trauma [18]. Abdominal malignancies and cirrhosis are the most common causes of chylous ascites in Western countries, in contrast, infectious etiologies (filariasis and tuberculosis) are responsible for the majority of cases in Eastern and developing countries. Abdominal malignancy is the most common etiological factor in adults, whereas congenital lymphatic abnormalities are more common in children [18,19].

The causes of chylous ascites also can be classified as traumatic and atraumatic [20]. Traumatic causes can be divided according to iatrogenic type, such as surgical, noniatrogenic, and idiopathic. Atraumatic causes can be divided according to whether they are neoplastic or congenital/acquired. Acquired diseases include cirrhosis, infectious, cardiac, gastrointestinal, inflammatory, and drug-related types [20]. Lymphatic anomalies, malignancy, liver cirrhosis, and mycobacterial infections have been reported to be the most frequent causes of atraumatic chylous ascites [21]. Chylous ascites has also been reported in patients using certain CCBs, especially patients undergoing PD [18]. CCBs/calcium antagonists are a heterogeneous group of drugs with diverse chemical structures and widely varying effects on cardiovascular function. They have been widely used for the treatment of hypertension and other cardiovascular conditions since the 1960s.

Chylous ascites caused by CCBs was first reported by Yoshimoto et al. in 1993, when manidipine was used [4]. Later in 1998, the same author found that other CCBs, including benidipine, nisoldipine, and nifedipine, had similar adverse effects [5]. Lercanidipine, azelnidipine, amlodipine, and diltiazem were reported to be associated with development of chylous ascites in 2006, 2010, 2011, and 2012, respectively.

The mechanisms of chylous ascites formation involve disruption of the normal lymph flow of the lymphatic system because of traumatic injury (surgery/trauma) or obstruction (from benign or malignant causes) [18,19]. Three underlying mechanisms of non-traumatic causes of chylous ascites have been proposed: (1) thoracic duct obstruction (primary or secondary), (2) mesenteric lymph gland obstruction (invariably secondary), and (3) congenitally incompetent megalymphatics [22]. However, the exact mechanism(s) of CCB-related chylous ascites is (are) still not established. Other possible factors include the following: (1) The lipophilic nature of CCBs—given that the newer CCBs are mostly lipophilic, investigators have suggested that these lipophilic CCBs contribute to the development of chylous ascites/effluents because of their unique action on the smooth muscle cells of the gastroenterological tracts and blood and lymphatic vessels [8,14]. (2) Decrease in lymphatic absorption—Hsiao et al. found that patients with lercanidipine-related cloudy effluent tended to have higher peritoneal membrane transport with an increased amount of effluent, which may lead to more lercanidipine accumulated in the peritoneal cavity through diffusion and, in turn, cause decreased lymphatic absorption [8]. (3) Continuous peritoneal lymph–vascular dilation—most authors suggest that CCBs may improve peritoneal vascular perfusion and therefore increase renal clearance or ultrafiltration through an effect on vascular smooth muscle cells [14]. (4) Increased lymphatic hydrostatic pressure—Basualdo et al. proposed that lercanidipine blocks the voltage-gated calcium channels present in lymphatic vessel smooth cells, resulting in a lack of contractility [23]. This interferes with lymphatic drainage, generates vasodilation, increases the hydrostatic pressure in lymph vessels, and causes exudation of lymph through the walls of dilated retroperitoneal vessels [2]. (5) Ethnic background—our study found that 46 of 48 patients with CCB-related chylous ascites were Asian and only two were European. Yang et al. hypothesized that different genetic or ethnic backgrounds may have different susceptibilities to chylous ascites. Moreover, individuals from the same ethnic group also have different susceptibilities. This may be because of polymorphisms in the calcium channel gene [14]. Our study also confirmed the Yang et al. study results showing that 46 of 48 cases were from Asia, with the other two from Spain (Europe).

The incidence of CCB-related chylous ascites varies. There have been different incidences in patients who had received different types of CCBs. Yoshimoto reported that 63% (five of eight) of PD patients taking manidipine developed chylous ascites [4]. The same author collected the experience of other nephrologists and found that 7.6% (19 of 251) PD patients experienced CCB-related chylous ascites [5]. Two retrospective studies conducted by Topal et al. and Yang et al. revealed that the incidence of lercanidipine-related chylous ascites was 13% and 57%, respectively [10,14]. Another prospective observation study reported by Hsiao et al. reported that the incidence of lercanidipine-related chylous ascites was 22.5% [8].

Chylous ascites frequently present as progressive and painless abdominal distension, which occurs over the course of weeks to months, depending on the underlying cause [20]. In our experience, CCB-related chylous ascites frequently presents as relatively mild or asymptomatic abdominal distension. The symptoms and/or chylous ascites appeared rapidly within 48 h (or 72 h) and disappeared within 24 h after withdrawal of the drugs. Rechallenge of CCBs in 10 patients produced recurrent chylous ascites, which also disappeared shortly after ceasing the drugs.

Abdominal paracentesis is the most important diagnostic tool in evaluating and managing patients with ascites. The triglyceride levels in ascitic fluid are very important in defining chylous ascites. Triglyceride values are typically >200 mg/dL [20,24], but some authors have used a cutoff value of 110 mg/dL [19]. The levels of triglycerides (TG) for the definition of chylous ascites were not present in all the cases of the present study because of variations. Some cases reported a TG level of >110 mg/dL, whereas others have >200 mg/dL. However, some cases have no definition of TG level for the diagnosis of chylous ascites including one case with TG = 81 mg/dL. This case is a manidipine hydrochloride-induced chyloperitoneum in a patient with systemic lupus erythematosus, and the reason for the low TG without PD is unknown. It has been suggested that mesenteric inflammation from lupus can lead to lymphatic obstruction and consequent chyloperitoneum and chylothorax. It may show a dilation effect because of an extensive field (thorax and abdomen). However, the effluent TG levels of the patients who received PD were significantly lower than the criteria of chylous ascites, probably because of the dilution effect by the indwelling of the PD volume [25].

Treatment of the underlying cause is an important initial step in managing patients with chylous ascites. If CCB-related chylous ascites is suggested, stopping the medication should be considered [25,26].

## 5. Conclusions

CCB-related chylous ascites is defined as white milky ascites/effluents that appear after administration of CCBs. Physicians must be aware of the possibility of chylous ascites when administering CCBs, particularly in patients with renal function impairment and ESRD patients undergoing PD. However, chylous ascites always resolves after cessation of the therapy, so awareness of this behavior can prevent unnecessary, expensive, and invasive diagnostic procedures and treatment.

## Figures and Tables

**Table 1 jcm-08-00466-t001:** Clinical and laboratory findings of all patients.

Year	Age (Years)/and Sex	Area	Drug	Received PD or Not	Effluent TG (mg/dL)	Time to Develop (h)	Time to Recovery (h)	Rechallenge Test	Reference
1993	39/2M, 3F	Japan	Manidipine	Yes	13,26,12,13,32 20 ± 9	24	<24	1 performed	[4]
1998	7M, 7F	Japan	Manidipine, Benidipine, Nisoldipine, Nifedipine	Yes	NM	48	NM (Shortly)	2 performed	[5]
1999	43F	Japan	Manidipine	No (renal dysfunction)	81	192	288	Not performed	[6]
2006	NM	Turkey	Lercanidipine	Yes	NM	24	NM	NM	[7]
2008	52.6 ± 18.5, 5M, 3F	Taiwan	Lercanidipine	Yes	128.4 ± 133.0	NM	<24	2 performed	[8]
2009	41/F	Taiwan	Lercanidipine	Yes	251	72	16 h	Reproducible	[9]
2010	39.4 ± 14.3 4M/5F	Taiwan	Lercanidipine	Yes	19.3 ± 6.3	28.8	<24	2 performed	[10]
2010	76/M	Japan	Azelnidipine	No (renal dysfunction)	NM	48	NM	NM	[11]
2012	55/M	India	Diltiazem	Yes	55.6	NM	<24	Reproducible	[12]
2014	59/F	Spain	Lercanidipine	Yes	20	72	24	Reproducible	[13]
2016	65/M	India	Amlodipine	Yes	293.8	72	<24	Not performed	[3]
2017	80/F	Spain	Lercanidipine	No (renal dysfunction)	321	48	NM	Not performed	[2]
2017	64/F	Turkey	Lercanidipine	Yes	80	24	24	Not performed	[14]
2017	82/M	Taiwan	Lercanidipine	Yes	NM	72	24	Not performed	Our patient
All	Mean:50								

NM = not mentioned; TG = triglyceride.

**Table 2 jcm-08-00466-t002:** The time of all patients to develop or recovery from chylous ascites/effluent after taking calcium channel blockers.

	<24 h	<48 h	<72 h
Time to develop (No./Total No.)	9/47 (19.1%)	34/47 (72.3%)	38/47 (80.1%)
Time to recovery (No./Total No.)	44/45 (97.8%)	44/45 (97.8%)	44/45 (97.8%)
Symptoms (No.)	Had symptoms (9)	No symptoms (24)	Not mentioned (15)

**Table 3 jcm-08-00466-t003:** Triglyceride (chylous ascites/effluent) levels of all patients.

	PD (Mean)	No PD (Mean)	Retrospective/Prospective Reports	Case Reports
Triglyceride (mg/dL)	74.1	201	59.1/70.6 *	80.1/140.1 *
Total cases	27	2	22/17 *	10/5 *

PD: Peritoneal dialysis. * *p* < 0.05.

**Table 4 jcm-08-00466-t004:** The lipophilicity of all calcium channel blocker-associated chylous ascites.

Classification	Drug	Case Number	Lipophilicity	Incidence	Percentage *
Dihydropyridine	Manidipine	16	5.6	41.7% (15/36)	33.3 (16/48)
	Benidipine	2	4.6	100% (2/2)	4.2% (2/48)
	Nisoldipine	1	3.3	9.1% (1/11)	2.1% (1/48)
	Nifedipine	1	2.2	0.6% (1/159)	2.1% (1/48)
	Azelnidipine	1	6.0	N	2.1% (1/48)
	Amlodipine	1	3.0	N	2.1% (1/48)
	Lercanidipine	25	6.9	24.7% (19/77)	52.1 (25/48)
Benzothiazepine	Diltiazem	1	3.1	N	2.1% (1/48)

N: Unknown. * Number/ Total number.

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
