# Peer review of "Calcium Channel Blocker-Related Chylous Ascites: A Systematic Review and Meta-Analysis"

_jcm, 2019, doi:10.3390/jcm8040466_

Round 1
Reviewer 1 Report
The article entitled "calcium channel blockers-related chylous ascites: A systematic review and meta-analysis", is interesting because describe an uncommon side effect of a very frequent used medications. However I have some major and minor comments and also suggestions for the authors.
Major Comments.
1. The leves of triglycerides necessary for the definition of chylous ascites was not present in all the cases of the present study. Although the authors suggested the PD as an important confounding and dilutional factor, one patient without PD also showed low TG count (81mg/dl). Authors should improve de description of the reasons why the PD could affect the measurements and why the patient described without PD, showed low TG leves.
2. In the previous context the authors should improve the description of the characteristics of the ascitic fluid in those cases in which the criteria for chylous ascites were not present. Like the serum-albumin gradient, leucocyte count and cultures.
3. The authors mentioned the term "meta-analysis" in the title, and although Its not my field of expertise, the volumen of the patients described, the amount of information derived and the type os statistical analysis probably were not consistent with a "meta analysis". Authors should obtain statistical support in this regard.
Minor comments:
1. In table 1, there are 2 same columns of "drug", and in the first row of the second an error of "menidipine". Authors should modify.
2. The authors described the potential block of CCB of the voltage dependent calcium channels in lymphatics. In this context, author should include the reference:
Lee S, Roizes S, von der Weid PV. Distinct roles of L-and T-type voltage -dependent Ca2 channels in regulation of lymphatic vessel contractile activity. J Physiol 2014;592(24):5409-5427
3. In reference 2: The title Percanidipine should be modified to Lercanidipine.
Author Response
The article entitled "calcium channel blockers-related chylous ascites: A systematic review and meta-analysis", is interesting because describe an uncommon side effect of a very frequent used medications. However I have some major and minor comments and also suggestions for the authors.
Major Comments.
1. The leves of triglycerides necessary for the definition of chylous ascites was not present in all the cases of the present study. Although the authors suggested the PD as an important confounding and dilutional factor, one patient without PD also showed low TG count (81mg/dl). Authors should improve de description of the reasons why the PD could affect the measurements and why the patient described without PD, showed low TG leves.
ANS: Thank you for your comments. The levels of triglycerides (TG) for the definition of chylous ascites were not present in all the cases of the present study because of variations. Some cases reported a TG level of >110 mg/dL, whereas others have >200 mg/dL. However, some cases have no definition of TG level for the diagnosis of chylous ascites including one case with TG = 81 mg/dL. This case is a manidipine hydrochloride-induced chyloperitoneum in a patient with systemic lupus erythematosus, and the reason for the low TG without PD is unknown. It has been suggested that mesenteric inflammation from lupus can lead to lymphatic obstruction and consequent chyloperitoneum and chylothorax. It may show a dilation effect because of an extensive field (thorax and abdomen). However, the effluent TG levels of the patients who received PD were significantly lower than the criteria of chylous ascites, probably because of the dilution effect by the indwelling of the PD volume. (see discussion red part on page 6).
2. In the previous context the authors should improve the description of the characteristics of the ascitic fluid in those cases in which the criteria for chylous ascites were not present. Like the serum-albumin gradient, leucocyte count and cultures.
ANS: Thank you for your comments. We have been added the description of the characteristics of the ascitic fluid in chylous ascites. (see discussion red part on page 5).
3. The authors mentioned the term "meta-analysis" in the title, and although Its not my field of expertise, the volumen of the patients described, the amount of information derived and the type os statistical analysis probably were not consistent with a "meta analysis". Authors should obtain statistical support in this regard.
ANS: Thank you for your comment! The term "meta-analysis" in the title was suggested by a statistician.
Minor comments:
1. In table 1, there are 2 same columns of "drug", and in the first row of the second an error of "menidipine". Authors should modify.
ANS: Thank you for your comments. We have been revised it. (see table 1 on page 3).
2. The authors described the potential block of CCB of the voltage dependent calcium channels in lymphatics. In this context, author should include the reference: Lee S, Roizes S, von der Weid PV. Distinct roles of L-and T-type voltage -dependent Ca2 channels in regulation of lymphatic vessel contractile activity. J Physiol 2014;592(24):5409-5427
ANS: Thank you for your comment! We have been added this reference in our manuscript. (see reference 23 on page 7).
3. In reference 2: The title Percanidipine should be modified to Lercanidipine.
ANS: Thank you for your comment! We have been revised it. (see reference 2 on page 7).

Reviewer 2 Report
Congratulations to authors reviewed CCBs. It would be also valuable if author could add some information whether any other medications may induce chylous ascites. In the literature review, Chylous ascites occurs after major abdominal surgery such as pancreatic surgery. How is the etiology differentiated if there were some cases developed chylous ascites after pancreatic surgery and taking CCB for HT.
Author Response
Congratulations to authors reviewed CCBs. It would be also valuable if author could add some information whether any other medications may induce chylous ascites. In the literature review, Chylous ascites occurs after major abdominal surgery such as pancreatic surgery. How is the etiology differentiated if there were some cases developed chylous ascites after pancreatic surgery and taking CCB for HT.
ANS: Thank you for your comments. Chylous ascites may be divided into traumatic and atraumatic causes. Surgical interventions (including post-pancreatic surgery) are well-known traumatic causes of chylous ascites secondary to direct lymphatic vessels injury. However, calcium channel blockers have also been implicated as an atraumatic cause of chylous ascites, and the exact mechanism involved is unclear. But most authors suggest that calcium channel blockers improve peritoneal vascular perfusion and therefore increase renal clearance through an effect on vascular smooth muscle cells.

Round 2
Reviewer 1 Report
Thanks to the authors for the answers and comments to the previous revision.